# An Assessment of Dam Operation Considering Flood and Low-Flow Control in the Han River Basin

**Jaewon Kwak**

Han River Flood Control Office, Ministry of Environment, Seoul 06501, Korea; firstsword@naver.com

**Abstract:** An assessment of dam operation is essential in dam management; however, there is a lack of a simple method that could be used in actual practice. This study aims for an actual dam operation evaluation method for flood and low-flow control of the three multi-purpose dams of Soyanggang, Chungju, and Hoengseong in the Han River basin, South Korea. Frequency matching method was applied to make a pair of cumulative distribution function (CDF) using daily dam inflow and outflow records. Runoff increasing and flood reduction rates are derived using CDFs of total and annual records. As a result, the average flood mitigation rates of the Chungju dam is approximately 35% annually and is relatively disadvantaged than the Soyanggang dam, which is 67.7% annually, due to small flood control capacity. The Hoengseong dam appeared to have a small flood reduction rate, but its runoff increasing rate is 94.7% annually because of the 209 km$^2$ upper basin area. The suggested method in this study could be used as a simple and intuitive field method to evaluate dam operations. Also, according to the annual evaluation, the Soyanggang and Chunju dam need more aggressive and anticipative operations for flood control such as pre-discharge before flooding or modify the Restricted Water Level (RWL) for flood seasons. On the other hand, Hoengseong dam need further data and studies.

**Keywords:** evaluation of dam operation; flood and low-flow control; frequency matching

## 1. Introduction

Flood and drought prevention and management, which are essential functions of a dam, are being magnified by climate change-induced disasters. This argument is especially clear during the 2014 to 2015 drought and the 2020 flood event in South Korea [1,2]. Despite the fact that assessment of dam operations is essential to conduct proper operation and enhance improvement, dam operation evaluation methods considering manipulation effects such as time delay or gate operation is still lacking.

There are many studies and methods being discussed for dam or reservoir operations. These methods are generally called as Reservoir Operation Method (hereafter referred to as "ROM"). Technical ROM, Spillway rule curve ROM, Rigid ROM, Auto ROM, Scheduled release discharge ROM, Linear decision rule, Stochastic Dynamic Programming, Standard Operation Policy, and Nonlinear Decision Rule are a few examples [3,4]. Other methods such as Ev-ROM take account the flood mitigation at the downstream of the dam [5], the reservoir operation criteria used to stabilize water supplies in a multipurpose dam [6], the reservoir operation method considering real-time prediction or operation [7,8], and operation strategy for ecological condition [9] are also suggested. Each method has its objective and needs valuable experience and insights of the experts because its operation was significantly influenced by many factors, including the storage amount, hydrological conditions, environmental constraints, and many others [2]. Also, dams can have a significant effect on river regimes according to their storing and operating policy [10]. In dam operation studies, the flow regime and its change are widely assessed because it is the first visible phenomenon of dam construction or operation. The indicator of hydrological alteration (IHA) [11] is a classic example of the assessment for flow regime alteration by

dam operation. Several indices to assess the flow regimes under dam constructions have been suggested [12–14], and its varying effects such as the organization of fish assemblages [15,16], biodiversity [17], riverbed adaptions [18], habitat suitability [19], water supply [20] are also assessed. On the other hand, some studies only focus on the evaluation of dam operation in flow regime [21,22]. Despite these studies, there are still difficulties in evaluating dam operation 'in actual practice for flood and drought' due to their complexities. Therefore, the actual evaluation is usually conducted based on the achievement rate relative to the operation plan, not on flood or drought [23].

The objective of this study is therefore to suggest a method that can quantitatively evaluate a dam or reservoir operation considering flood and low-flow control and its actual application to evaluate multipurpose dams in the Han River basin. The historical record of the dam inflow and outflow for the Soyanggang, Chungju, and Hoengseong dam are obtained. The flood reduction and runoff increasing rates, which are represented as the ratio of runoff volumes at the flood and low-flow water level, are derived using the frequency matching method and its proper truncation level. The evaluation of dam operation was conducted and analyzed for the multipurpose dam in the Han River basin.

## 2. Methodologies

### 2.1. Frequency Matching

The frequency matching, which is the method to match a frequency of two data series, is widely used to adjust the systematic error or sampling due to their ease of use. In hydrologic field, it also often used to correct the bias of forecasting rainfall [24–26], or to estimate the parameter of radar rainfall [27], and can also be applied to derive the concept of the Curve Number (hereafter referred as "CN") [28]. The frequency matching was conducted by independently sorting in ascending (or descending) order and by combining the data that has the same frequency into a pair. While these new "ordered" pairs may not occur in nature, each data has the same frequency or return period. This is based on the assumption that the cause and its results have the same frequency, for example, the 100-year flood is estimated from the 100-year rainfall (Figure 1), and it is a major use of the CN method [28].

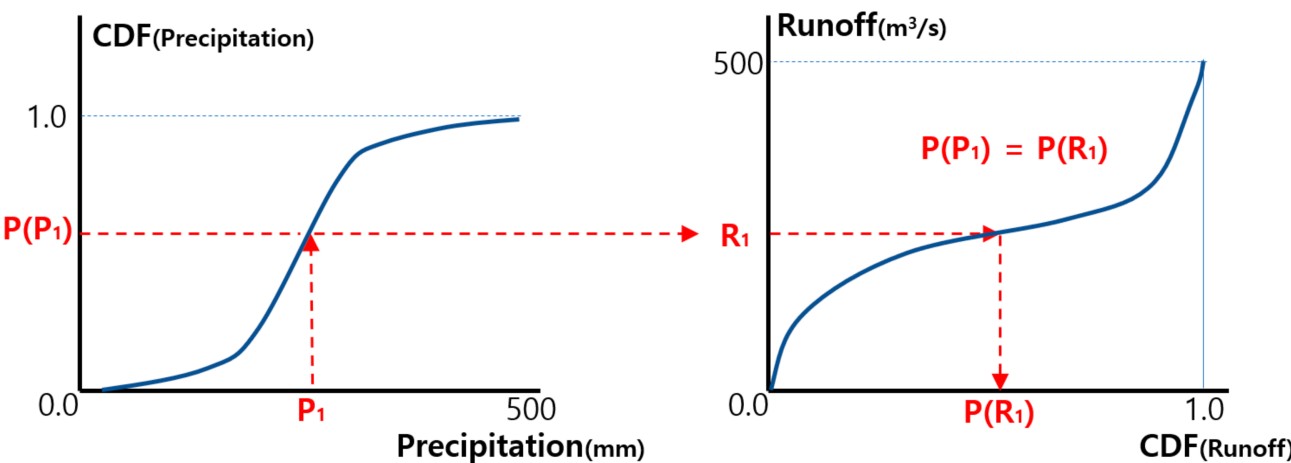

**Figure 1.** Concept of frequency matching in the CN method.

### 2.2. Evaluation Method of Dam Operation

In terms of disaster prevention of the dam, it can be summarized into two primary ways: flood reduction and runoff increase. During the low-flow period, the dam outflow is promoted using stored water in the dam, while during flooding, a large portion of the inflow was kept in the dam, therefore only a small portion is being discharged. These operations result in changes in the outflow trend itself. The runoff volumes are increased due to additional discharge at low-flow, and are decreased at flooding (Figure 2a). Despite

being occasionally evaluated using the change of runoff volumes, challenges due to the time-delay between in- and outflow, and as well as varying delays in disaster events can occur.

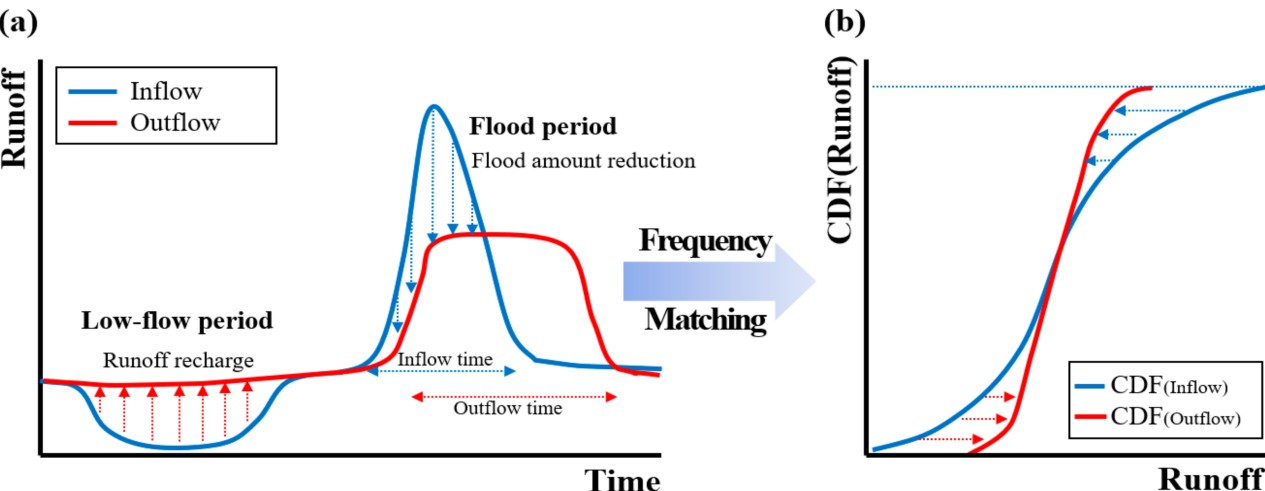

**Figure 2.** Dam operation and its result in CDF; (**a**) concept of dam operation; (**b**) the change in CDF due to dam operation.

The concept of the frequency matching method would be a good alternative. According to the frequency matching, the inflow amount with a 100-year return period will cause the outflow amount that has the same frequency. In the same vein, there are differences in frequency between in- and outflow amounts due to the dam operation, including inflow, storage, and release of water. Without the effect from hydrostatic pressures, human operations is a significant cause of these difference. Therefore, inflow and outflow series are expressed as a pair of the cumulative probability distribution (CDF) using the frequency matching method (Figure 2b), and the difference between them is a result of dam operation.

If inflow and outflow series correspond to CDF of the same frequency, the result of flood and low-flow control can be found at both tails of CDFs. The flood reduction and runoff increase could be estimated in terms of the ratio of runoff volumes on both sides of the CDF for flood and low-flow areas. However, regular dam operations are conducted to secure or control the storage amount without being a flood and low-flow control. Therefore, it is necessary to set a specific truncation level of low-flow ($T_{low-flow}$) or flood ($T_{flood}$) to exclude ordinary dam operations. Through this, the flood reduction rate ($\alpha_{flood}$) at flooding and the runoff increasing rates ($\propto_{low-flow}$) at low-flow periods could be estimated as shown in Figure 3 and Equation (1).

$$\propto_{low-flow} = \left(1 - \frac{\int_{R_{I1}}^{R_{I2}} (P(R_i) - P(R_{i-1}))\ R\ dR}{\int_{R_{D1}}^{R_{D2}} (P(R_i) - P(R_{i-1}))\ R\ dR}\right)$$
$$\propto_{flood} = \left(1 - \frac{\int_{R_{D3}}^{R_{D4}} (P(R_i) - P(R_{i-1}))\ R\ dR}{\int_{R_{I3}}^{R_{I4}} (P(R_i) - P(R_{i-1}))\ R\ dR}\right)$$

(1)

where, $R_{Di=1,4}$, $R_{Ii=1,4}$ are the start and end points of each CDF, respectively and $R_{Di=2,3}$, $R_{Ii=2,3}$ are the points that correspond to truncation level of flood and low-flow, respectively. $P(R_i)$ and $P(R_{i-1})$ are the probability of each runoff value $R_i$ and $R_{i-1}$. Therefore, each side of the fraction in Equation (1) could be expressed as the product of the area of the probability density function and runoff amount. When both sides of the fraction in Equation (1) shows the same value, which means the same outflow correspond to inflow amount, each rate ($\alpha_{flood}$, $\propto_{low-flow}$) will be calculated as zero. Otherwise, each rate will show a higher value when the upper side of the fraction is large. Therefore, each flood reduction and runoff increasing rate (%) could be defined as the ratio of water stored or supplied on the dam relative to the inflow amount. Also, the truncation level of flood and

low-flow to exclude ordinary dam operations may adapt the existing regulations. The standard design of multi-purpose dams is supposed to withstand drought with 20-year return periods [29] which is a suitable truncation level of low-flow. Similar standards may be applicable to flood truncation level, but the dam operation in flooding events varies on dam's structure, hydrological conditions, and many others. Therefore, it is deemed necessary to select a proper truncation level for each dam.

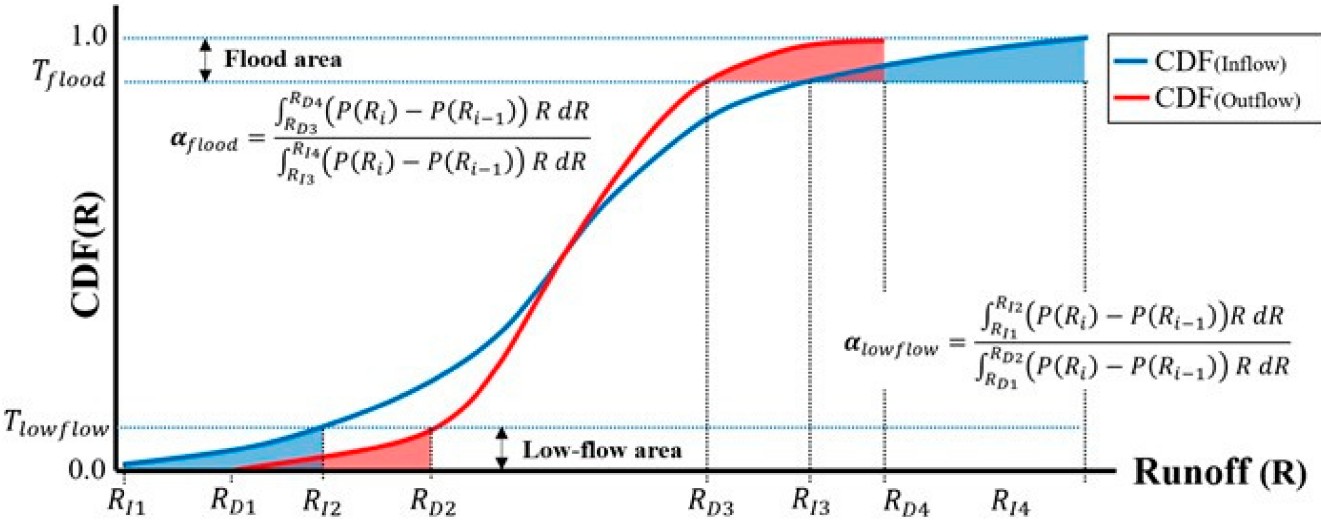

**Figure 3.** The basic concept of evaluation for dam operation; colored area indicates the range of each area of flood and low-flow.

## 3. Application and Discussion

### 3.1. Study Material

The target of the study are the three multipurpose dams in the Han River basin; the Soyanggang, Chungju, and Hoengseong dam, which are located in the Namhan River, Bukhan River, and Seom River which are the main tributaries of the Han River (Figure 4). The Soyanggang dams have 2703 km$^2$ of upper basin area, 2.9 billion m$^3$ of storage capacity, 500 million m$^3$ of flood control volumes, and 1.2 billion m$^3$ of water supply amount annually. On the other hand, Chungju dams have 6648 km$^2$ of upper basin area, 2.8 billion m$^3$ of storage capacity, 616 million m$^3$ of flood control volumes, and 3.3 billion m$^3$ of water supply amount annually. The Soyanggang and Chungju dam are located in the mainstream of the Bukhan River and Namhan River, respectively. The Hoengseong dam is located in the upper part of the Seom River and has 209 km$^2$ of upper basin area, 86.9 million m$^3$ of storage amount, 9.5 million m$^3$ of flood control volumes, and 119.5 million m$^3$ of water supply amount annually [30].

88% of the Soyanggang dam, 92% of Chungju dam, and 90% of Hoengseong dam basins are covered by deciduous, coniferous, and mixed forest [31]. Also, all dams have several control gates, emergency spillways, an outlet of hydropower, as well as the restricted water level (R.W.L) for flood season. So, it does not need to consider hydrostatics pressure on outflow amount since manual operations determine it. All of them have an important role in the water supply and flood mitigations of the Han River due to the large difference in precipitation during regular and monsoon periods. Thus, the major objectives of these dams are flood mitigations for monsoon and maintain water supply and storage inflow amount for other periods. The historical records of inflow (Figure 5) and outflow on these dams are obtained from the Water Resources Management Information Systems [31] and the Hoengseong and Wonju Office of the K-Water corporation [32].

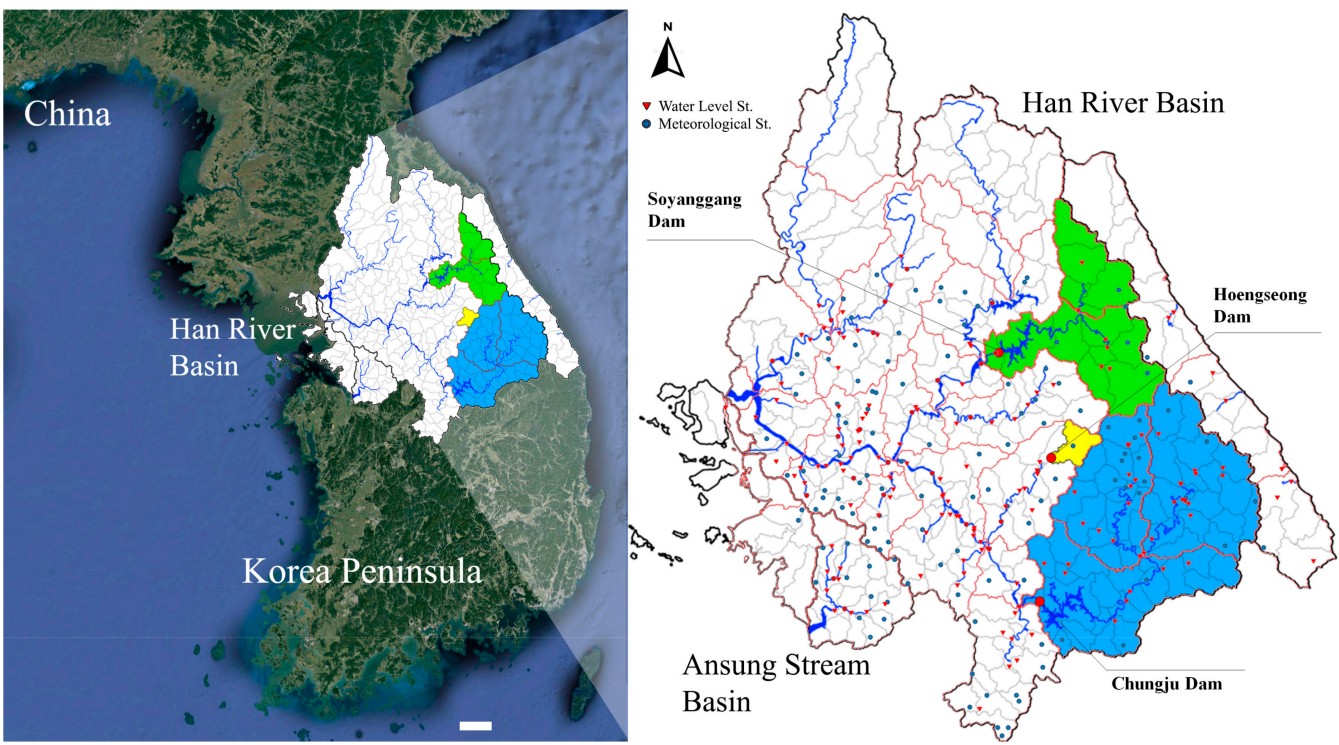

**Figure 4.** Study Area and Dam; red circle indicate the location of each dams, and colored area indicate that the upper basin area of each dam.

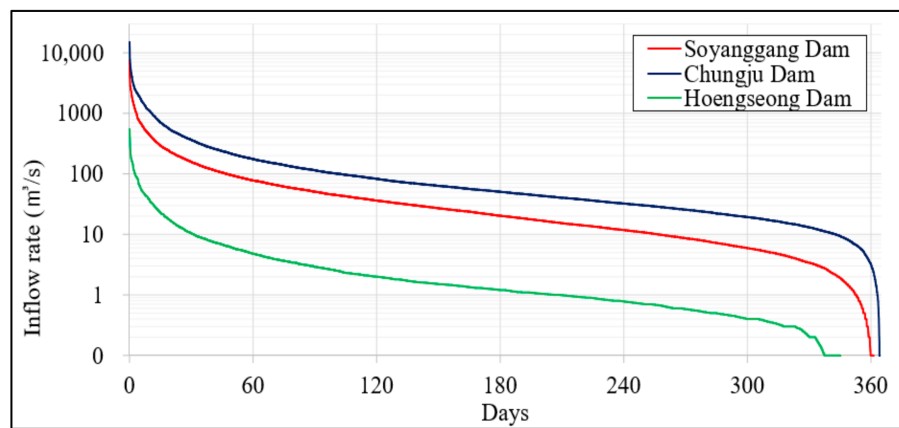

**Figure 5.** Study Area and Dam; red dot indicate the location of each dams, and colored area indicate the upper basin area of each dam.

### 3.2. The Evaluation of Dam Operations for Flood and Low-Flow Control

The frequency matching method are applied to historical records of inflow and outflow on the multipurpose dams: the Soyanggang dam for 1976 to 2020, the Chungju dam for 1987 to 2020, and the Hoengseong dams for 2001 to 2020. Also, the truncation level for flood and low-flow should be determined to exclude an ordinary dam operation. For low-flow, the inflow rate of 20-year return periods ($P(R) = 0.05$), which are stipulated in the standard for dam design, was used as the truncation level for low-flow. There are no clear criteria for threshold values of inflow rates for flood mitigation so the same criteria for low-flow, which are described in the 20-year return period ($P(R) = 0.95$), are used (Table 1).

**Table 1.** Truncation level of flood and low-flow level on each dam.

| Truncation Level of Inflow | Inflow Rate of Dam (m³/s) | | |
|---|---|---|---|
| | Soyanggang | Chungju | Hoengseong |
| Food $P(R) = 0.95$ (20 yr return period) | 250.7 | 594.0 | 19.5 |
| Median | 20.0 | 49.7 | 1.3 |
| Low-flow $P(R) = 0.05$ (20 yr return period) | 2.0 | 9.0 | 0.2 |
| Historical median for Oct. 2014 to Jun. 2015 | 8.2 | 32.1 | 0.7 |

The truncation level of low-flow were estimated as 2.0, 9.0, and 0.2 m³/s for the Soyanggang, Chungju, and Hoengseong dam, respectively. The inflow rates were compared to the median values during 2014 to 2015 which is the most severe drought for the last 30 years [32]. Upon comparison, the truncation level is found to be suitable. Also, there is a 250 m³/s flow rate for the Soyanggang dam and 594 m³/s flow rate for the Chungju dam after a heavy rainfall event. However, 19.5 m³/s flow rate is relatively low for flooding on the Hoengseong dam. Approximately, a 25.0 m³/s flow rate is expected during gate operations for flood mitigations according to the Hoengseong and Wonju office of K-Water, the agencies mainly in-charge of the dam [33]. Its level corresponds to 25-year return period ($P(R) = 0.96$), and these truncation levels are shown in Figure 6.

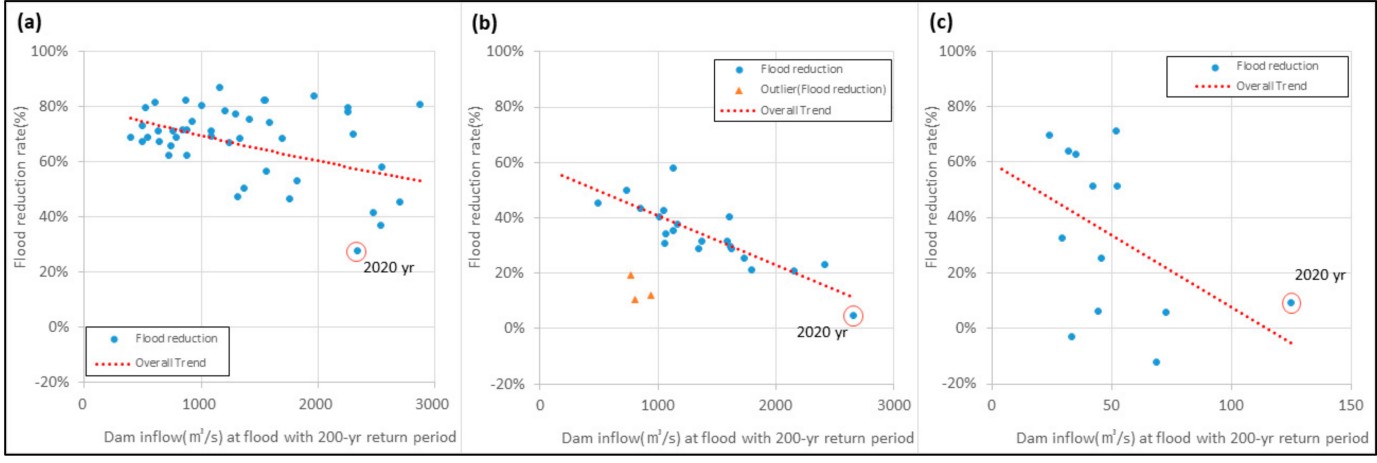

**Figure 6.** CDF of each dam record using frequency matching; (**a**) Soyanggang dam for 1976 onwards; (**b**) Chungju dam for 1987 onwards; (**c**) Hoengseong dam for 2001 onwards; red box indicate calculated area based on truncation level for flood and low-flow.

Using a pair of CDF with frequency matching method in Figure 6 and Equation (1), the runoff increasing and flood reduction rate (%) are derived and shown in Table 2. The Soyanggang dam has shown 72.6% of flood reduction rate during flooding, and 65.7% of runoff increasing rate at low-flow. On the other hand, Chungju dam has shown a 53.3% of flood reduction and 77.6% of runoff increasing. Overall, the Soyanggang and Chungju dam shows relatively similar rates (60~70%) of runoff increasing rate, and it means that both of them seem to contribute to the runoff conditions in the downstream area of the dams. But, in terms of flood reduction, there is a 20% difference between them. This could have been because of the upper basin area and storage capacity of the dams. The volume of flood mitigations in the Chungju dam is 616 million m³ and it is approximately 1.2 times higher than the Soyanggang dam (500 million m³), but the upper basin area is 6648 km² and it is approximately 2.5 times higher (2703 km² of the Soyanggang dam). Therefore, the flood mitigation function of the Chungju dam is relatively disadvantaged, and that could be the reason why the flood reduction rate is low. Unlike the other multipurpose dams, the Hoengseong dam has a relatively small capacity for storage and flood mitigations and showed small flood reduction rates of 15.9%. However, the runoff increasing rate

of the Hoengseong dam is 97.4%, the highest among others. Also, the 209 km² of the upper basin area seems to contribute due to the small amount of baseflow, while the runoff increasing functions of the Hoengseong dam contributes significantly to the Seom River. Based on the evaluation result, the Chungju dam needs more aggressive plan for flood mitigations. The operation of the Hoengseong dam should consider both runoff as well as other environmental conditions.

**Table 2.** Flood reduction and runoff increasing rates for each dam.

| Dam Operation | Reduction or Increasing Rate (%) | | |
| --- | --- | --- | --- |
| | Soyanggang | Chungju | Hoengseong |
| Flood reduction | 72.6 | 53.3 | 15.9 |
| Runoff increasing | 65.7 | 77.6 | 97.4 |

*3.3. Annual Evaluation of Dam Operation and Discussion*

The evaluation of dam operation for flood and low-flow were conducted based on the frequency matching method and the historical records of the multipurpose dam in the Han River basin. However, annual evaluation and further improvement are also essential to achieve proper operations and improvement for the dam. It would be conducted according to the same method in Section 3.2 using an annual historical record of the dam. The truncation level for the flooding for whole periods could be applied for annual evaluation, but the level for low-flow for whole periods cannot be applied since hydrological or runoff conditions are different every year. Therefore, the truncation level for low-flow were separately estimated each year which corresponds to the 20-year return period ($P(R) = 0.05$). Also, the flow regime of each year significantly influenced the dam operation and its evaluation. There may be no low-flow period when the flow regimes are higher than the normal year, and there may also be no flood events according to the flow regime. Therefore, to consider flow regimes of each year, the annual flow rate at low-flow with 20-year return period ($P(R) = 0.05$), and at flood with 200-year return periods ($P(R) = 0.995$) are also considered. The annual evaluation result of dam operation for flood and low-flow and flow regimes are shown in Figures 7 and 8.

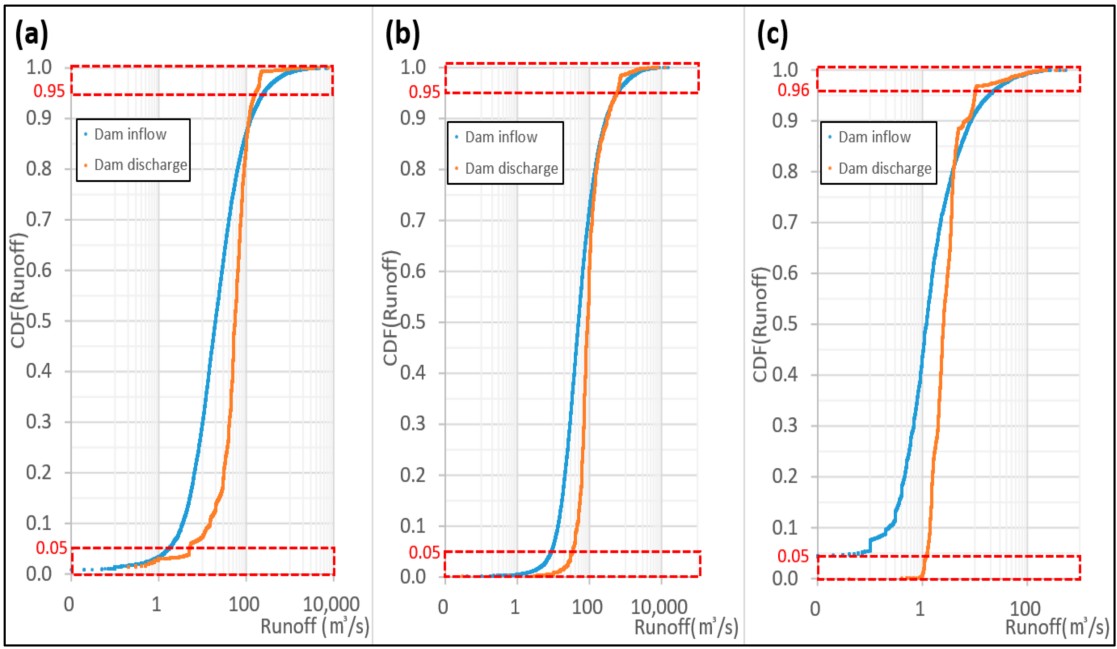

**Figure 7.** Scatter diagram of runoff increasing rate vs. dam inflow at low-flow level with 20-year return period; (**a**) Soyanggang dam; (**b**) Chungju dam; (**c**) Hoengseong dam; red circle indicate record's year which are severe drought during last 30 years.

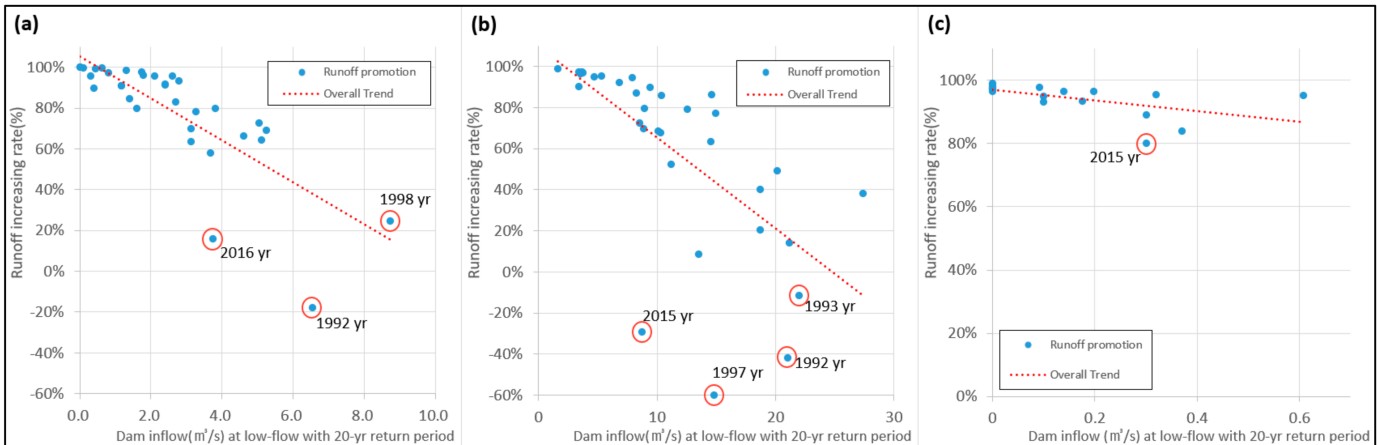

**Figure 8.** Scatter diagram of flood reduction rate vs. dam inflow at flood level with 200-year return period; (**a**) Soyanggang dam; (**b**) Chungju dam; (**c**) Hoengseong dam; red circle indicates the 2020 record which is one of the most severe flood events during the last 30 years.

All of the multipurpose dams show a decreasing trend for the runoff increasing as low-flow rate at 20-year return period increase (Figure 7), (Table 3). That supports the fact that the dam outflow shows a similar amount as the inflow because it does not have to discharge additional flow when the flow regime is better than normal. The average runoff increasing rate of the Soyanggang dam is 67.7% annually and decreased to 40% or less according to inflow rate at low-flow with 20-year return period. The Chungju dam shows 35% of rates annually and it also decreased to 20% according to the low-flow with 20-year return period. Both dams show some negative cases in which a fewer outflow volume were recorded compared to the inflow volumes. It could be the result of the extreme drought countermeasure, which have cut down the water supply and the river maintenance flow. Both dams showed these negative rates in 1992 to 1993, 1997 to 1998, and 2015 to 2016 which are the lowest storage amount recorded during severe drought [31]. Unlike the others, the Hoengseong dam consistently shows a 90% or more runoff increase rate even during the most severe drought event in 2015 to 2016. The Hoengseong dam was completed in 2000 and was not affected by previous severe drought events, but it was able to have more than 90% or runoff increase rate during 2015. This is because the upper basin area of the Hoengseong dam is 209 km$^2$ only. It means that the inflow rate at low-flow is extremely low, therefore the water supply amounts are either equivalent or more than the inflow. As a result, the Soyanggang and Chungju dam will be able to operate flexibly, but the Hoengseong dam will be needing more storage amount to maintain the flow regime of the Seom River.

**Table 3.** Annual flood reduction and runoff increase rates for each dam.

| Dam Operation | | Reduction or Increasing Rate (%) | | |
|---|---|---|---|---|
| | | Soyanggang | Chungju | Hoengseong |
| Flood reduction | average rate | 67.7 | 35.0 | 33.4 |
| | Std. deviation | 14.0 | 8.6 | 29.9 |
| Runoff increasing | average rate | 79.4 | 60.6 | 94.7 |
| | Std. deviation | 26.7 | 43.9 | 5.0 |

Similar to the evaluation results of low-flow, the flood reduction rate tends to decrease when the flow rate corresponding to the 200-year return period has increased (see Figure 8 and Table 3). In the case of the Chungju dam (Figure 8b), those in 1989, 2000, and 2001 were located in the southwest part and has a different trend than the others. However, they have 700 to 900 m$^3$/s of flow rate corresponding to the 200-year return

period. It is a relatively small amount than others so all of them were considered to be outliers and therefore ignored. Generally, the dam should open the gate frequently when the flow regime with 200-year return period is higher than normal. In the Soyanggang dam, average flood reduction rate is 67.7% annually and its variability seems to be large than the Chungju dam. It decreases to about 40% when the flow rate at 200-year return period increases. The Chungju dam shows a similar trend, but it has a more clear and consistent decreasing trend with 35.0% flood reduction rate annually. As described earlier, it seems due to the characteristics of the dam. The flood control capacity of the Chungju dam is 616 million m$^3$ which is relatively small relative to 6648 km$^2$ of the basin area; therefore the flood control capacity could be decreasing rapidly when there is high flooding. These tendencies were repeated during flooding, showing the trend in Figure 8b. Therefore, the Chungju dam operation needs more anticipative countermeasure or capacity for flood, such as pre-discharge before flooding, and establishing more flood control capacity to secure the same level of the flood reduction with the Soyanggang dam. Using Figure 7,8, annual evaluation of dam operation, which consider flood and low-flow control, would be capable and it could be used as the elementary standards for field engineers. However, for the Hoengseong dam, it seems that there are significant influence from the storage conditions or flooding amount, but there are relatively small data periods (2001 to 2020) and the total 9-yr record (2001, 2006, 2008, 2014~2016, 2018~2019) does not top the truncation level of flooding. This means that it needs more data and studies to estimate the relationship between them.

Also, there is an important point that could not be neglected, which is the dam operation in 2020. Both dams show a lower rate of flood reductions relative to previous years, just 28% for the Soyanggang dam and only 5% for the Chungju dam. A total of 652.6 mm and 512.3 mm of monthly precipitation in August on the upper basin of the Soyanggang and Chungju dam, respectively, were recorded. These records are equivalent to nearly half of yearly precipitation. Approximately 1030 mm of precipitation during two months (15 July to 15 September), which corresponds to 81% of the annual total (Figure 9), and the relatively high storage amount in the dam both helps in the water supply and seem to cause the lower rates. Therefore, more aggressive and anticipative operations for flood countermeasures such as pre-discharge before flooding or modification of the Restricted Water Level (RWL) for flood seasons are necessary to secure additional flood control capacity.

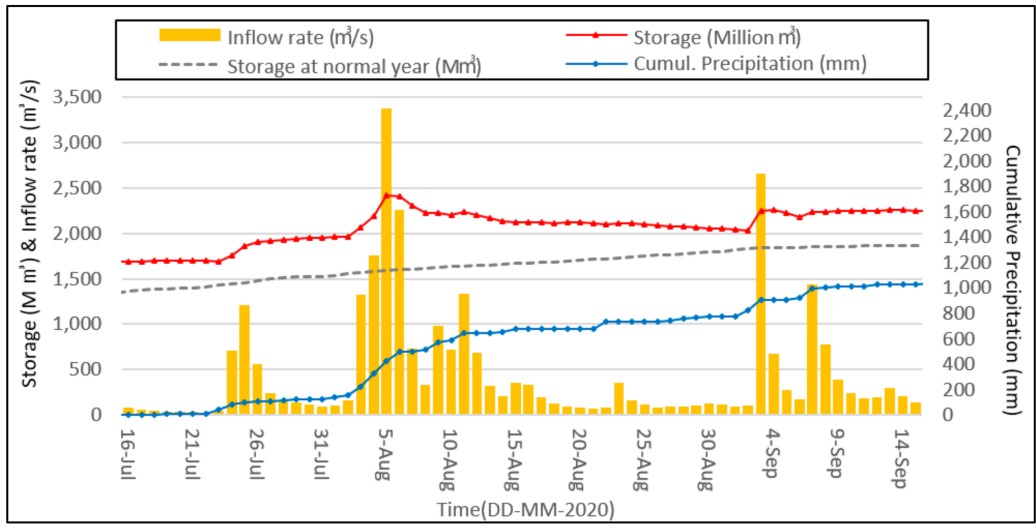

**Figure 9.** Precipitation, storage amount, and inflow rates during 2020 monsoon season (15 July to 15 September).

This study is limited to the suggested method and index. When a drought is forecasted to continue, dam discharge may be reduced to ensure water supply even though it is in

low-flow periods. In the same way, dam discharge during flooding events may also be increased in order to secure the flood control capacity for a forecasted heavy rainfall event. But since the suggested method and index could not come up with a good strategy, an offset to some degree by annual or total periods of data can be done. This is why this study used annual or whole periods instead of events. Another one is the characteristics of the dam, all of the studied dams has several control gates, emergency spillway, and outlet of hydropower. Therefore, hydrostatics pressure on outflow amount is no longer necessary since it could be determined manually. However, small dams often do not have a discharge structure and will be affected by hydrostatic pressure. The outflow amount is also inevitably affected, and it means that the suggested method of the study is difficult to apply. Thus, further studies that take account the hydrostatic pressures are needed.

## 4. Conclusions

The objective of this study is to establish the method to evaluate dam operations of flood and low-flow control and apply it to the three multi-purpose dams in the Han River basin, South Korea. Daily dam inflow and outflow records are obtained for the dams, and the frequency matching method was applied to make a pair of CDFs. The dam operation on the flood and low-flow were separated based on the truncation level ($P(R) = 0.05$, $P(R) = 0.995$). Finally, the runoff increase ($\propto_{low-flow}$) and flood reduction ($\alpha_{flood}$) rates (%), which can be defined as the ratio of water stored or supplied on the dam relative to the inflow amount, were derived.

These evaluation rates are estimated and analyzed for the Soyanggang, Chungju, and Hoengseong dam. The Soyanggang dam resulted to a 72.6% flood reduction and 65.7% runoff increasing rate, and the Chungju dam resulted to have 53.3% flood reduction and 77.6% runoff increasing rates for whole periods. The flood mitigation function of the Chungju dam is relatively disadvantaged than the Soyanggang dam due to the small flood control capacity relative to the upper basin area. The Hoengseong dam has relatively small storage and flood control capacity, and it appeared to have a small flood reduction rate of 15.9%, but its runoff increasing rate is 97.4% because of the 209 km$^2$ upper basin area.

Also, an annual evaluation of dam operation was conducted to identify the degree of proper dam operations and the points that need improvement. Annual flow rate at low-flow with 20-year return period and at flood with 200-year return period are also considered for flow regimes of each year. All of the multipurpose dams show a decreasing trend in runoff increase rate according to the increase in low-flow rate at 20-year return period. The Soyanggang and Chungju dam shows 79.4% and 60.6% average annual runoff increase rate, respectively. The Hoengseong dam shows 90% or more rates and needs enough storage amount to maintain a flow regime of the Seom River. Similar to the low-flow, the flood reduction rate tends to decrease when the flow rate corresponding to the 200-year return period increases. The Soyanggang dam has 67.7% of flood reduction rates annually, although its variability is higher than the Chungju dam. The Chungju dam has a 35.0% flood reduction rates annually, and shows a similar trend, but it has a clearer and more consistent decreasing trend due to its relatively small flood control capacity. Also, it seems that the flood control capacity decreases rapidly when there are high floods in the dam. Therefore, the Chungju dam operation needs more anticipative countermeasures or capacity for flood. For the Hoengseong dam, further data and studies seem to be needed. Also, the Soyanggang and Chunju dam show a lower rate of flood reductions compared to the previous years. Heavy rainfall in 2020 and relatively high storage amount in the dam seem to be the cause of these lower rates. Therefore, more aggressive and anticipative operations for flood countermeasure such as pre-discharge before flooding or modification of the Restricted Water Level (R.W.L.) for flood seasons are necessary to secure additional flood control capacity. Indeed, the suggested methodologies in this study could be used as an intuitive and quantitative method to evaluate dam operations.

**Funding:** This research received no external funding.

**Institutional Review Board Statement:** Not applicable.

**Informed Consent Statement:** Not applicable.

**Data Availability Statement:** Available on request from corresponding author.

**Acknowledgments:** I would like to thank to the Flood Forecast and Control Division of Han Flood Control Office and the Center for Hydrology and Ecology of INHA University.

**Conflicts of Interest:** The authors declare no conflict of interest.

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
