# Peer review of "An Assessment of Dam Operation Considering Flood and Low-Flow Control in the Han River Basin"

_water, doi:10.3390/w13050733_

Round 1

Reviewer 1 Report

Paper gives practical insight into the very common engineering problem of the spillway flow out of the dam. Although the manuscript is prepared very well, I am suggesting a major revision. Here are my thoughts.

1) Subchapter 3.1. Location map should also contain a wide view where a case study position could be seen.

2) Authors should describe how flow amounts with respect to the probability occurrence were calculated.

3) Figures 6 and 7 present key results of the analysis. Defining a linear trend, as well as any trend in this case is not recommended. Better solution is applying a Taylors' diagram instead of defining trends. Here the linear trending does not have justification.

4) Although this is a paper written in a form of application of the scientific methodology for real case study for the purpose of giving a forecast for the value of flow amount, authors should take into account that hydrostatic pressure on a dam also defines a amount of the water, which should be released. Authors should comment on this. 

Author Response

Paper gives practical insight into the very common engineering problem of the spillway flow out of the dam. Although the manuscript is prepared very well, I am suggesting a major revision. Here are my thoughts.

1) Subchapter 3.1. Location map should also contain a wide view where a case study position could be seen.

-> figure 4 was revised to show the wide topography of the study area.

2) Authors should describe how flow amounts with respect to the probability occurrence were calculated.

-> The description to the calculation of each rate were revised in section 2.2

3) Figures 6 and 7 present key results of the analysis. Defining a linear trend, as well as any trend in this case is not recommended. Better solution is applying a Taylors' diagram instead of defining trends. Here the linear trending does not have justification.

-> I politely ask to give another comment for Taylor’s diagram. Taylor’s diagram is an effective tool to identify which of the models have an explanation power. However, the index that was suggested in the study is not a model or an approximated representation, it is just a ratio of water stored or supplied to the dam relative to the inflow amount. So, it should not be compared to the correlation coefficients, standard deviations, and RMSE. For instance, it needs a large difference between inflow and outflow amount to show higher rates of flood reduction or runoff promotion. It also means that there are small values of the correlation coefficients, and large values of the standard deviations and RMSE. Indeed, the Taylor’s diagram for the three dams is the same as below, and it’s hard to find clear trend.

Figure 1. Taylor’s Diagram for the Soyanggang Dam

Figure 2. Taylor’s Diagram for the Chungju Dam

Figure 3. Taylor’s Diagram for the Hoengseong Dam

I agree to the reviewer’s comments.  There are small meanings for regression due to large variability. It would be more accurate to describe “the regression of each rates” as “overall trend of each rates”. Also, another important thing is the ease of use. The study also aims to generate a method for actual practice that will include clear and simple calculation even by just using MS EXCEL. However, the Taylor’s diagram is a more complex method when it comes to actual practice.

But ofcourse, I will follow the reviewer’s comments. Again, I politely ask the reviewer to give a comment regarding the use of Taylor’s diagram.

4) Although this is a paper written in a form of application of the scientific methodology for real case study for the purpose of giving a forecast for the value of flow amount, authors should take into account that hydrostatic pressure on a dam also defines a amount of the water, which should be released. Authors should comment on this. 

-> I appreciate you’re the reviewer’s keenly comments. It is the one of the important constraints but I have missed it. Consequently, all of studied dams have several gates, emergency spillway, and hydropower outlet which are fully controlled. Also, all of them has R.W.L (Restricted Water Level) for flood mitigations. Therefore, hydrostatic pressures are barely affected by the outflow amount. But definitely, it is one of the important constraints that can determine the outflow amount. This point was discussed in the manuscript in section 3.1 and conclusion.

Reviewer 2 Report

Dear Author,

Ms. Ref. No.: water-1123917

Title: An Assessment of Dam Operation considering Flood and Low flow Control in the Han River Basin

Journal: Water

General Comments

The manuscript deals with the assessment of dam operation by using cumulative distributive frequency distribution curves for low flows and high flows events in three multipurpose dams.

The author have done a good job by analyzing these relationships as dam operations control are one of the demanding topic in the hydrological world. In addition, by doing so it will help the stakeholder and policy makers to better predict the events beforehand. However, despite all these, there are many major concerns in this manuscript, which has to be undertaken before considering it for publication in this journal. As the author have mentioned the dam is used for multipurpose operations, so it would be great if they can show the allocation of different purposes and landuse map of the area charactering how the runoff is affected based on different landuse patterns. In addition, it is not clear from the results how the dam operates for monsoon period and non-monsoon period, explain more about the CDF and perhaps include flow duration curves FDCs analysis, that can be found in these important studies mention below:

Zhang, Y., Shao, Q., & Zhao, T. (2017). Comprehensive assessment of dam impacts on flow regimes with consideration of interannual variations. Journal of Hydrology, 552, 447-459.

Wang, H., Lei, X., Yan, D., Wang, X., Wu, S., Yin, Z., & Wan, W. (2018). An ecologically oriented operation strategy for a multi-reservoir system: A case study of the middle and lower Han River Basin, China. Engineering, 4(5), 627-634.

I will also recommend author to include the line numbers in the manuscript. As it becomes difficult to locate the issues without line numbers.

Introduction section is poorly written and organized without describing the previous literature and research gaps. Restructuring of entire Introduction is required. Objectives of the work does not show any novel aspect which is due to the fact that author have missed the important aspects in this field. The author have discussed about the low flows and high flows however, the methodology used in this classification is not clear. I suggest going through the recent literature to update the methodology for classification.

Proper care of English language should be taken, as the manuscript is not written well. There are many sentences, which are formed incorrectly. Many grammatical errors – far more than is appropriate for a scientific reviewer to make edits on, but these need to be addressed before publication.

Specific comments

There is a line spacing in between the abstract. Novel aspect in the abstract section is missing as only future suggestions are given. There should be a concrete findings stated in the abstract.

Grammar issues not limited to this:

Introduction:

First line of the Introduction: Please check the sentence it is grammatically incorrect. Again there is usage of “Generally speaking” which is not used in scientific writing try to avoid these type of phrases.

“However, there are lack” – are should be replaced by is.

There is a huge English problem all over the manuscript. Extensive editing is required to improve the quality of the manuscript.

The first of statement of the second paragraph is entirely wrong and not supported by any citation.

SRC, SRD, LDR, SDP, etc should be in bracket nor its full form.

Literature review is very poorly written and discussed. Relevant and recent literature is missing from first two paragraphs. First author is mentioning the citation in number then jumped to alphabets (A1, A2……..) is there any specific reason behind that.

Method:

Frequency matching technique is not advance one so how do author justify with other studies that this technique is unique.

Citation for curve number is wrong please correct it.

The assumption made in the section 2.1 should go to the Introduction section

‘and it is a major use of the CN method’ please revise this sentence

Figure 1 -  x-axis is not mentioned and expand “precip” to precipitation.  Expand y-axis clearly and elaborate the caption of this figure.

‘it also cause challenges since because of the time-delay between in- and outflow’ – revise this sentence

‘The concept of the frequency matching method would be a good alternative to overcome the aforementioned challenges since it states that The change in outflow volume due to the inflow with 100-year return period will have the same frequency’. – This sentence is not at all appropriate with the context and should go in the introduction.

The major concern in the method is usage of CN method – how do you account CN values based on your landuse characteristics. Provide a detailed calculation and steps for the calculation of equivalent CN values. Also, what type of soil is accounted and depending on that how the antecedent soil moisture conditions are taken.

There should be a table describing the general charestireretics of the all stations in the catchment. For example: topographic features, landuse, weather, reservoir capacity.

In Figure 4 please show all the meteorological and gauging station locations

In section 3.2 what is the criteria to partition low flows. Have the author used any mathematical approach to support this criterion.

Also in this section, it has been mentioned that Chungju dam has received heavy rainfall events, so please show a time series plot of precipitation and streamflow for clear visualization of the events.

Results:

Interannual variations of dam regulation on entire flow regime are not shown that can be important to see the seasonal patterns.

In order to predict the flows accurately and justify with the observed ones, it is recommended using some statistical analysis. For example, multivariate statistical analysis is one of the tool to determine the multicollinealrity among flow regimes metrics and the dimensionality curse, which are serious concerns when analyzing the impact of dam regulation.

Figure 6: its mention in y-axis runoff promotion rate and in the caption runoff increasing rate. It is required to be uniform. I am sorry to say but there is not any new information from this graph which can be insightful I recommend to do some further analysis as mentioned above.

Discussion is entirely missing, results section English is too bad to understand. Please revise there are a lot of redundancy in the sentences, which needs to be corrected. I would recommend to use abovementioned literature to support the findings of this study again listed below:

Zhang, Y., Shao, Q., & Zhao, T. (2017). Comprehensive assessment of dam impacts on flow regimes with consideration of interannual variations. Journal of Hydrology, 552, 447-459.

Wang, H., Lei, X., Yan, D., Wang, X., Wu, S., Yin, Z., & Wan, W. (2018). An ecologically oriented operation strategy for a multi-reservoir system: A case study of the middle and lower Han River Basin, China. Engineering, 4(5), 627-634.

The statement ‘In the Soyanggang dam, the variability of the flood reduction rate seems to be large’ is not been seen in any of the figures shown in the manuscript. Have the author described any index such as coefficient of variation to depict such variation.

Conclusion:

No need to explain CDF again.

The font size is not uniform throughout the manuscript. For example, P(R) = 0.05 is different than P(R) = 0.995

Author Response

The manuscript deals with the assessment of dam operation by using cumulative distributive frequency distribution curves for low flows and high flows events in three multipurpose dams.

The author have done a good job by analyzing these relationships as dam operations control are one of the demanding topic in the hydrological world. In addition, by doing so it will help the stakeholder and policy makers to better predict the events beforehand. However, despite all these, there are many major concerns in this manuscript, which has to be undertaken before considering it for publication in this journal.

As the author have mentioned the dam is used for multipurpose operations, so it would be great if they can show the allocation of different purposes and land use map of the area charactering how the runoff is affected based on different landuse patterns. In addition, it is not clear from the results how the dam operates for monsoon period and non-monsoon period, explain more about the CDF and perhaps include flow duration curves FDCs analysis, that can be found in these important studies mention below:

Zhang, Y., Shao, Q., & Zhao, T. (2017). Comprehensive assessment of dam impacts on flow regimes with consideration of interannual variations. Journal of Hydrology, 552, 447-459.

 Wang, H., Lei, X., Yan, D., Wang, X., Wu, S., Yin, Z., & Wan, W. (2018). An ecologically oriented operation strategy for a multi-reservoir system A case study of the middle and lower Han River Basin, China. Engineering, 4(5), 627-634.

-> First, Thank you for the reviewer’s keenly comments. Land use and FDCs are important in  distinguishing basin characteristics including the runoff amount. Therefore, the FDCs of each dam are added in the manuscript.

However, the land use map does not fit the manuscript because most of them are covered by forest which is approximately 90%. Therefore, discussion about Land use can be found in section 3.1. The Land use (cover) map is the same as below:

 I will also recommend author to include the line numbers in the manuscript. As it becomes difficult to locate the issues without line numbers.

-> The manuscript was submitted according to the word template of the Water Journal. I am not sure if I can modify the template. But if that is permissible, I will add the line number in the manuscript.

Introduction section is poorly written and organized without describing the previous literature and research gaps. Restructuring of entire Introduction is required. Objectives of the work does not show any novel aspect which is due to the fact that author have missed the important aspects in this field. The author have discussed about the low flows and high flows however, the methodology used in this classification is not clear. I suggest going through the recent literature to update the methodology for classification.

-> Introduction section was revised.

Proper care of English language should be taken, as the manuscript is not written well. There are many sentences, which are formed incorrectly. Many grammatical errors – far more than is appropriate for a scientific reviewer to make edits on, but these need to be addressed before publication.

-> Grammar was revised.

Specific comments

There is a line spacing in between the abstract. Novel aspect in the abstract section is missing as only future suggestions are given. There should be a concrete findings stated in the abstract.

Grammar issues not limited to this:

-> I think there is some editing issue. It was revised.

-> Abstract was revised.

Introduction:

 First line of the Introduction: Please check the sentence it is grammatically incorrect. Again there is usage of “Generally speaking” which is not used in scientific writing try to avoid these type of phrases.

 “However, there are lack” – are should be replaced by is.

  • These sentences were revised.

There is a huge English problem all over the manuscript. Extensive editing is required to improve the quality of the manuscript.

 The first of statement of the second paragraph is entirely wrong and not supported by any citation.

 SRC, SRD, LDR, SDP, etc should be in bracket nor its full form.

-> Introduction section was revised, and first sentence was removed. All abbreviations were changed to full form.

Literature review is very poorly written and discussed. Relevant and recent literature is missing from first two paragraphs. First author is mentioning the citation in number then jumped to alphabets (A1, A2……..) is there any specific reason behind that.

-> Literature review are revised, and jumped number is just a mistake in editing and it was also revised.

Method:

Frequency matching technique is not advance one so how do author justify with other studies that this technique is unique.

 Citation for curve number is wrong please correct it.

 The assumption made in the section 2.1 should go to the Introduction section

 ‘and it is a major use of the CN method’ please revise this sentence

-> It’s right, the frequency matching is not a new or an advanced technique at all. But it doesn’t mean that the method could not be used. The application in practical field is a reason. During the last 5 years, there are significant damage from flood and low-flow in Asia including China, Korea, and Japan, and these disasters aggravate due to climate change. A major concern of the author is that there is a lack of ‘field technique’ for dam evaluation in actual practice. This is the reason why we used frequency matching. The study also aims ease of use, so we do not suggest a completely new technique, but suggest a technique that can be easily applied in the field. Indeed, the method of the study could be conducted with daily in- and outflow record and MS Excel. I politely ask the reviewer to perceive this study in the perspective of the practical users.

-> Citation for the curve number and several sentences were revised, and the assumption was mentioned in the introduction section.

Figure 1 -  x-axis is not mentioned and expand “precip” to precipitation.  Expand y-axis clearly and elaborate the caption of this figure.

  • Figure 1 was revised

‘it also cause challenges since because of the time-delay between in- and outflow’ – revise this sentence

  • That sentence was revised.

 ‘The concept of the frequency matching method would be a good alternative to overcome the aforementioned challenges since it states that The change in outflow volume due to the inflow with 100-year return period will have the same frequency’. – This sentence is not at all appropriate with the context and should go in the introduction.

The major concern in the method is usage of CN method – how do you account CN values based on your landuse characteristics. Provide a detailed calculation and steps for the calculation of equivalent CN values. Also, what type of soil is accounted and depending on that how does the antecedent soil moisture conditions are taken.

-> First, my apologies for the inclusion of these paragraphs. I think the sentence doesn't convey the meaning very well. CN method was not used in the study, just the concept of the frequency matching are used. So, section 2.2 was revised.

There should be a table describing the general charestireretics of the all stations in the catchment. For example: topographic features, landuse, weather, reservoir capacity.

-> Overall characteristics are discussed in section 3.1. But the land use map does not fit the manuscript because most of the area is covered by forest at approximately 90%. So, Land use are mentioned in section 3.1. Land use (cover) map is the same as below:

In Figure 4 please show all the meteorological and gauging station locations

-> Figure 4 was revised to show water level (including runoff) and meteorological stations.

In section 3.2 what is the criteria to partition low flows. Have the author used any mathematical approach to support this criterion.

-> The truncation level for low-flow is that is corresponding to a 20-year return period. In fact, there is no clearly mathematical criteria on where is the low-flow. But, in South Korea, the standard design of multi-purpose dams is supposed to withstand drought with 20-year return period. So, this study employed the same criteria which is on a 20-year return period.

Also in this section, it has been mentioned that Chungju dam has received heavy rainfall events, so please show a time series plot of precipitation and streamflow for clear visualization of the events.

-> Figure 9 was added in the discussion section. But, I would like to ask the reviewers if I can remove Figure 9. Approximately, 80% of the yearly total are pouring in just 2 months, and it seems to be enough to describe these lower rate. Most important thing is, these rates could be used as a measure of dam operation in the perspective of flood and low-flow mitigations. Therefore, I think it relatively do not fit this manuscript. I politely ask the reviewer to give another comment regarding this issue.

Results:

Interannual variations of dam regulation on entire flow regime are not shown that can be important to see the seasonal patterns.

In order to predict the flows accurately and justify with the observed ones, it is recommended using some statistical analysis. For example, multivariate statistical analysis is one of the tool to determine the multicollinealrity among flow regimes metrics and the dimensionality curse, which are serious concerns when analyzing the impact of dam regulation.

-> I agree with the reviewer’s comment thatthe challenge in flow regime such as multicollinealrity is important to predict runoff. Multivariate statistical analysis or principal component analysis is one of the alternatives for these challenges. However, this study aims to evaluate the dam operation in the perspective of flood and low-flow mitigations, not predict in- or outflow amount. These kinds of analyses do not match the study. However, to follow the the reviewer’s comments, the Discussion section was revised and table 3 was added in the manuscript.

Figure 6: its mention in y-axis runoff promotion rate and in the caption runoff increasing rate. It is required to be uniform. I am sorry to say but there is not any new information from this graph which can be insightful I recommend to do some further analysis as mentioned above.

-> Figure 7 was revised.

Discussion is entirely missing, results section English is too bad to understand. Please revise there are a lot of redundancy in the sentences, which needs to be corrected. I would recommend to use abovementioned literature to support the findings of this study again listed below:

Zhang, Y., Shao, Q., & Zhao, T. (2017). Comprehensive assessment of dam impacts on flow regimes with consideration of interannual variations. Journal of Hydrology, 552, 447-459.

 Wang, H., Lei, X., Yan, D., Wang, X., Wu, S., Yin, Z., & Wan, W. (2018). An ecologically oriented operation strategy for a multi-reservoir system: A case study of the middle and lower Han River Basin, China. Engineering, 4(5), 627-634.

 The statement ‘In the Soyanggang dam, the variability of the flood reduction rate seems to be large’ is not been seen in any of the figures shown in the manuscript. Have the author described any index such as coefficient of variation to depict such variation.

-> Discussion section was revised and table 3 was added in the manuscript.

Conclusion:

No need to explain CDF again.

The font size is not uniform throughout the manuscript. For example, P(R) = 0.05 is different than P(R) = 0.995

-> These sentence was revised.

Round 2

Reviewer 1 Report

Authors provided very quality improvement of the manuscript. All answers to my comment are justified, as well as elaborated in detail. I am strongly suggesting publishing of the paper. 

Reviewer 2 Report

The authors have addressed all of the comments adequately therefore I suggest for publication.